# Data Leakage Detection and De-duplication in Large Scale Geospatial Image Datasets

## Abstract

In our study, we conducted a comprehensive analysis of three widely used datasets in the domain of building footprint extraction using deep neural networks: the INRIA Aerial Image Labelling dataset, SpaceNet 2: Building Detection v2, and the AICrowd Mapping Challenge datasets. Our experiments revealed several issues in the AICrowd Mapping Challenge dataset, where nearly 90% (about 250k) of the training split images had identical copies, indicating a high level of duplicate data. Additionally, we found that approximately 56k of the 60k images in the validation split were also present in the training split, amounting to a 93% data leakage.

Furthermore, we present a data validation pipeline to address these issues of duplication and data leakage, which hinder the performance of models trained on such datasets. Employing perceptual hashing techniques, this pipeline is designed for efficient de-duplication and leakage identification. It aims to thoroughly evaluate the quality of datasets before their use, thereby ensuring the reliability and robustness of the trained models.

## 1 Introduction

In recent years, deep learning and pattern recognition techniques have had a significant impact on remote sensing. In particular, a number of works have employed popular CNN architectures such as UNets and ResNets (Ronneberger et al., 2015; He et al., 2016; Chatterjee & Poullis, 2019; Zorzi et al., 2021; Girard et al., 2021; Li et al., 2021a; Xu et al., 2023) as well as attention-based architectures (Li et al., 2019; Zorzi et al., 2022; Hu et al., 2023) for tasks such as building footprint extraction, road network extraction, etc., which have important applications in downstream urban understanding tasks such as land use and land cover classification, urban planning, navigation, etc. Deep learning solutions that can generalize to unseen data distributions require an abundance of data, which has a significant impact on their applicability. Hence, the availability of large-scale, high-resolution remote sensing image datasets is crucial for the success of such methods. In light of this, it is imperative to assess the quality of such datasets to determine their suitability for developing such deep-learning solutions.

**Benchmark Datasets:** Owing to the need for large-scale image datasets of high quality, the majority of deep learning literature tends to adopt widely used publicly available benchmark datasets to train and evaluate their methods and compare with existing state-of-the-art works. For building footprint extraction, the need for high-quality, curated datasets containing polygonal building footprints has prompted a significant amount of recent literature to utilize the AICrowd mapping challenge dataset (Mohanty et al., 2020) extensively for training and testing their methods, as well as for comparison with other state-of-the-art methods. Other popular datasets include the INRIA Aerial Image Labelling Dataset (Maggiori et al., 2017) and the SpaceNet Building Detection dataset (Etten et al., 2018), however, these datasets either only provide raster building mask annotations or provide data in a non-standard format (e.g., GeoJSON, GeoTIFFs) for the computer vision and deep learning research community. The AICrowd dataset claims to solve this problem by providing large-scale, high-resolution satellite images with polygonal building footprint annotations made available in the popular MS-COCO format (Lin et al., 2014), allowing the immediate use of this dataset by the computer vision research community. Consequently, many recent works addressing the task of polygonal building footprint extraction have evaluated their methods

using the AICrowd mapping challenge dataset, either in conjunction with other datasets or exclusively.

**State-of-the-art trained on the AICrowd Mapping Challenge dataset:** Our experiments on the three datasets and subsequent analyses, as explained later in the paper, revealed significant issues with the AICrowd Mapping Challenge dataset. Furthermore, several recent studies (Li et al., 2019; Zhang & Aliaga, 2022; Zorzi et al., 2021; Lee et al., 2021; Girard et al., 2021; Xu et al., 2023; Hu et al., 2023; Wei et al., 2023) that focus on the task of building footprint extraction from remotely sensed imagery have used this contaminated AICrowd Mapping Challenge dataset in their experiments to evaluate their proposed methods. In some studies (Zhao et al., 2020; Wang & Zhang, 2022; Zorzi et al., 2022), this dataset has even been used *exclusively* to benchmark and evaluate the proposed methods. Upon reviewing these works, it is evident that the AICrowd dataset has been extensively used in recent literature, which further motivates us to evaluate the quality of this dataset and inform the research community of the several issues discovered in this dataset.

**Impact of Dataset Quality on Deep Learning Models:** The issues discovered in the AICrowd dataset, and our subsequent analyses of methods using the dataset, make it clear that contamination in large image datasets negatively impacts the reusability, robustness, and generalization of models trained on such datasets. Excessive duplication and leakage in a benchmark dataset often lead to the trained models exhibiting overfitting behavior and performing poorly on out-of-distribution data at test time. Issues like data leakage also have implications for the fairness and reliability of machine learning benchmarks commonly used by the research community to evaluate ongoing research efforts. Therefore, such datasets must be carefully analyzed for issues such as data leakage, excessive duplication, etc., before being used for model training/evaluations. However, this has become increasingly difficult to perform for large image datasets such as the ImageNet dataset (Deng et al., 2009), MS-COCO dataset (Lin et al., 2014), the Cityscapes datasets (Cordts et al., 2016), etc., which can have up to several million image-annotation pairs, and also newer datasets such as the LAION-5B dataset (Schuhmann et al., 2022) that can even have billions of image-annotation pairs. Therefore, there is an imperative need to develop efficient methods to evaluate and mitigate dataset quality issues (such as duplication and data leakage) on such large image datasets.

**De-duplication of Large Image Datasets:** Recent research has focused on de-duplicating large image datasets using neural network feature representations of images to detect duplicates. In CE-Dedup (Li et al., 2021c), the authors use a hashing-based image de-duplication technique to significantly reduce the size of the dataset while still maintaining the accuracy of downstream image classification tasks. In Jafari O. et al (Jafari et al., 2021), the authors study the suitability of locality-based hashing in a variety of downstream applications, such as machine learning and image/video processing. The authors in (Li et al., 2021b) present QHash, a hashing algorithm for image de-duplication in datasets containing images with small visual differences, such as medical images.

In contrast to the hashing-based approaches described above, more recent methods also employ self-supervised pretraining schemes to learn image descriptors that are then used in identifying similar images in the dataset (Pizzi et al., 2022; Zhang et al., 2023). Although such pretraining approaches could potentially achieve a higher degree of de-duplication for specific datasets, pretraining on very large datasets can be challenging and may not generalize well to other substantially different datasets. In light of these recent works, we adopt a perceptual hashing strategy to investigate the degree of data duplication and leakage in the AICrowd dataset.

**Motivation and Contributions:** The INRIA Aerial Image Labelling and SpaceNet 2: Building Detection v2 datasets were observed to indicate no major issues upon scrutiny. However, the same cannot be said for the AICrowd Mapping Challenge dataset.

Due to its significant size and the availability of building footprint annotations, numerous state-of-the-art methods have employed the AICrowd dataset extensively for training and validation (Zorzi et al., 2021; Girard et al., 2021; Li et al., 2021a; Xu et al., 2023; Li et al., 2019; Zorzi et al., 2022; Hu et al., 2023). However, a careful examination of this dataset reveals a plethora of issues. These include duplication and data leakage across officially provided splits. These issues have a considerable impact on the performance of downstream applications where this dataset is used for training and evaluating building footprint extraction methods.

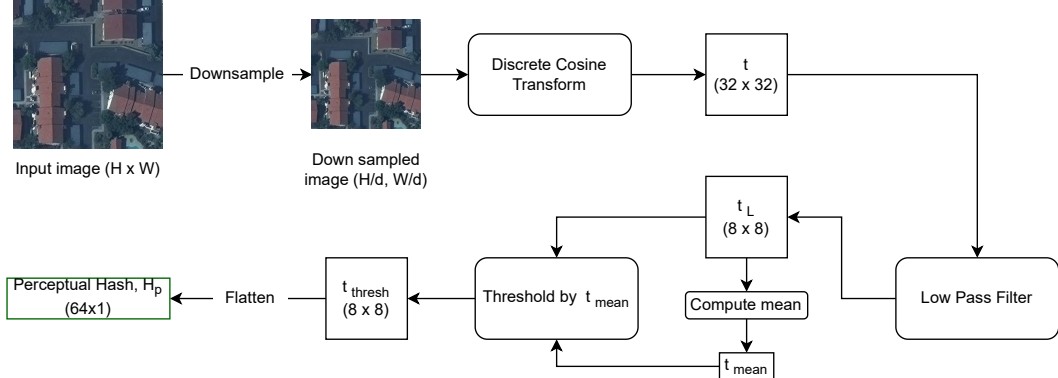

Figure 1: The pipeline used for computing the perceptual hash, $H_p$ of an image. The input image is first downsampled by a downsampling factor, $d$. The $32 \times 32$ discrete cosine transform, $t$ of the downsampled image is computed and the lowest frequencies, $t_L$ (top-left $8 \times 8$ values) are retained. Finally, $t_L$ is thresholded by the mean of the retained low frequencies and flattened to result in the $64 - d$ perceptual hash, $H_p$ of the input image.

This underlines the impetus for our study: the demand for an effective, easy-to-adopt pipeline to swiftly evaluate the quality of large-scale image datasets. Such pipelines could conserve the time and effort of the research community, allowing for more efficient use of available resources. Specifically, our contributions are as follows:

- Our study presents a thorough analysis of three key large-scale remote-sensing datasets, with a particular focus on the AICrowd Mapping Challenge dataset. In this in-depth analysis, we identify and highlight critical issues such as extensive duplication, where nearly 89% of the training images are duplicates (either exact or augmented), and significant data leakage, with about 97% of the validation images also present in the training split.
- Complementing our analytical findings, we present a deduplication and leakage detection pipeline, specifically tailored for large-scale image datasets. By utilizing perceptual hashing methods to detect collisions, this pipeline is a practical and easy-to-adopt method for identifying and eliminating data duplication and leakage issues. Its application in analyzing the INRIA Aerial Image Labelling dataset, SpaceNet 2: Building Detection v2, and particularly the AICrowd Mapping Challenge, demonstrates its practicality and efficacy in enhancing dataset integrity, thereby contributing to improving the robustness of machine learning pipelines.

## 2 METHODS

We present an effective method for detecting and eliminating data duplication and leakage in large-scale image datasets. The pipeline is independent of any particular dataset and it is based on the calculation of perceptual hashes of images in the dataset, as shown in Figure 1. This method efficiently identifies exact duplicates as well as augmented copies of images in a dataset. Augmented copies are images that have undergone transformations such as rotations and flips but remain inherently the same image.

**Datasets:** We have evaluated three popular benchmark datasets widely used for training deep neural networks on building and building footprint segmentation: INRIA Aerial Image Labelling dataset (Maggiori et al., 2017), SpaceNet 2: Building Detection v2 (Etten et al., 2018), and AICrowd Mapping Challenge (Mohanty et al., 2020).

The **INRIA Aerial Image Labelling Dataset** (Maggiori et al., 2017) consists of public domain aerial images and building footprint masks of size $5000 \times 5000$ with a spatial resolution of 0.3m. The official train split consists of 180 such tiles with corresponding binary ground truth building masks. The official test split consists of another set of 180 images whose annotations are not publicly available. In our experiments, we split each image in the dataset into 250px $\times$ 250px non-overlapping

patches to result in 72,000 patches in the train split and 72,000 in the test split. These splits were used in our de-duplication and leakage detection experiments.

The **SpaceNet 2: Building Detection v2 dataset** (Etten et al., 2018) consists of 24,586 satellite image scenes across four areas (Las Vegas, Paris, Shanghai, and Khartoum). The images are of size 650px × 650px with a spatial resolution of 0.3m. The officially provided train and test splits were used in our de-duplication and leakage detection experiments.

The **AICrowd Mapping Challenge dataset** (Mohanty et al., 2020), derived from the larger SpaceNet v2 challenge dataset, is composed of $300 \times 300$ RGB patches of WorldView 3 satellite images, each with a spatial resolution of 0.3m. The dataset is reasonably large, with 280,741 images in the training set and 60,317 images in the validation set. All images include MS-COCO annotations of polygonal building footprints (Lin et al., 2014).

We apply our method to the above datasets. The INRIA Aerial Image Labelling Dataset (Maggiori et al., 2017) and the SpaceNet 2: Building Detection v2 dataset (Etten et al., 2018) passed the scrutiny of our pipeline without revealing any major issues. These datasets consist of aerial and satellite images, respectively, with spatial resolutions of 0.3m, and are used in their officially provided train and test splits. However, when applying our pipeline to the AICrowd Mapping Challenge dataset, we discovered several issues as discussed in the next subsection.

## 2.1 DATA LEAKAGE & DE-DUPLICATION

To illustrate the applicability of the proposed method and subsequent de-duplication, we use the AICrowd Mapping Challenge dataset as a representative example. Below we describe the steps taken to analyze and address the issues identified.

**Initial Observations.** To determine the scope of data leakage between the official training and validation splits of the AICrowd dataset, we calculated the perceptual hashes, as shown in Figure 1 of the images in the official training and validation splits and also their corresponding augmented versions.

| Search Set | Targets Set | Total number of images in Search set | Number of targets in Search set | Overlap % |
|---|---|---|---|---|
| Official Train | Official Train | 280,741 | 166,193 | 59.19% |
| | Official Val | | 95,241 | 33.92% |
| | Official Test | | 95,884 | 34.15% |
| Official Val | Official Train | 60,317 | 56,368 | **93.45%** |
| | Official Val | | 10,658 | 17.67% |
| | Official Test | | 20,642 | 34.22% |
| Official Test | Official Train | 60,697 | 56,608 | **93.26%** |
| | Official Val | | 20,639 | 34.00% |
| | Official Test | | 10,701 | 17.63% |
| Augmented Train | Augmented Train | 280,741 | 251,403 | 89.55% |
| | Augmented Val | 1,684,446 | 642,741 | 38.16% |
| | Augmented Test | 1,684,446 | 645,205 | 38.30% |
| Augmented Val | Augmented Train | 361,902 | 351,182 | **97.04%** |
| | Augmented Val | 60,317 | 46,151 | 76.51% |
| | Augmented Test | 361,902 | 139,330 | 38.49% |
| Augmented Test | Augmented Train | 364,182 | 353,041 | **96.94%** |
| | Augmented Val | 364,182 | 139,349 | 38.26% |
| | Augmented Test | 60,697 | 46,483 | 76.58% |

Table 1: **Data Leakage and Duplication.** Summary of the extent of data leakage/duplication in the train, validation, and test splits of the AICrowd dataset. The 'Official' train/val/test sets are those provided by the AICrowd Mapping Challenge dataset (Mohanty et al., 2020). The 'Augmented' train/val/test sets refer to those obtained after augmenting the official sets with 90°, 180°, 270° rotations, and horizontal and vertical flips.

After computing the hashes for all images in the official splits provided in the AICrowd dataset, we checked for exact duplicates across the splits by searching for exact hash collisions. We observed significant data leakage between the official training and validation splits of the AICrowd dataset. The results of these comparisons are displayed in Table 1 with some examples illustrated in Figure 2.

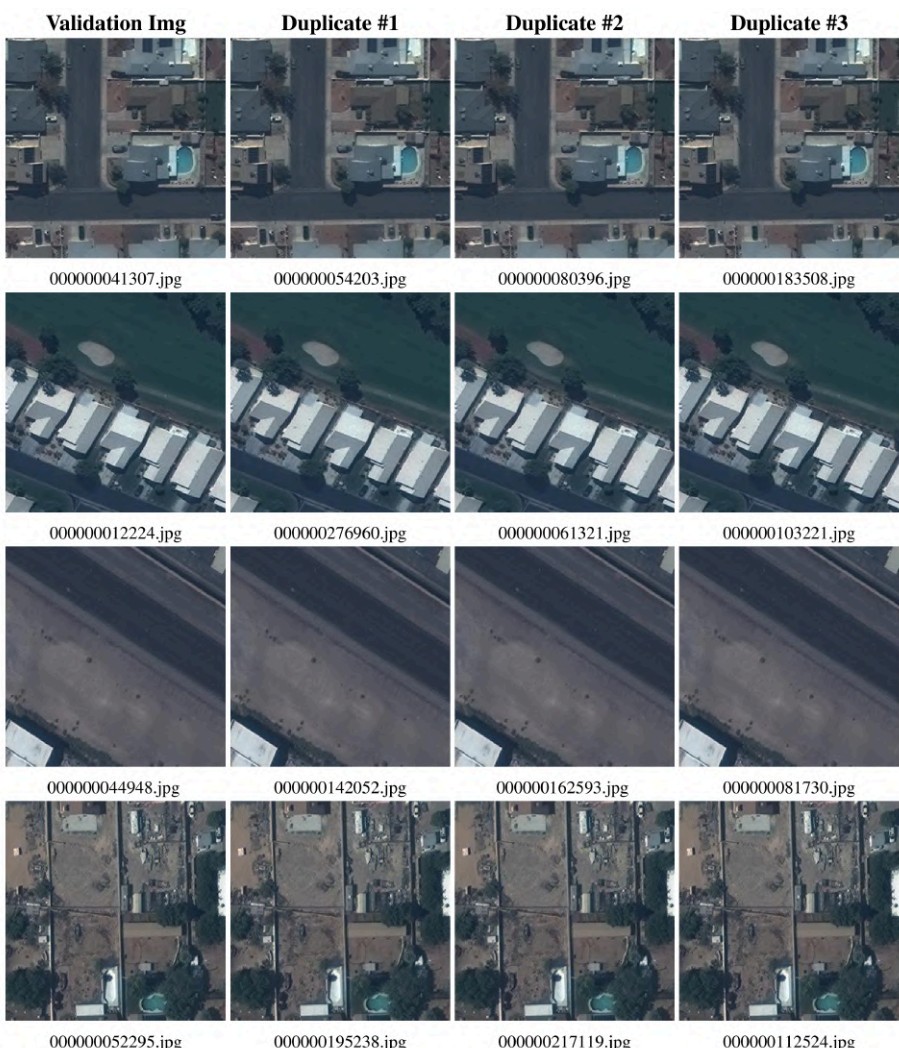

Figure 2: **Data Leakage.** Here we show examples of data leakage in the AICrowd dataset (Mohanty et al., 2020) (CC BY-NC-SA 4.0). We sample four images from the **validation split** and show duplicates occurring in the **training split**.

**Leakage Detection in the Official Splits.** Further analysis showed that several additional images in the training split were augmented copies of images in the validation split. To detect such augmented duplicates, we augmented every image in the validation split with the following augmentation: $90°$, $180°$, $270°$ rotations, and horizontal and vertical flips. The perceptual hashes of this augmented validation set were then compared to that of the training images. In this case, we found that $38.72\%$ (108,707) of the official training images were exact or augmented duplicates of images found in the official validation split. Based on these findings, it is evident that a significant portion of the validation split appears multiple times in the training split of the AICrowd dataset, resulting in significant data leakage.

**Eliminating Augmented Duplicates.** The following procedure was adopted to address the issue of data leakage between the official training and validation splits of the AICrowd dataset. First, we augmented all images in the official train split with $90°$, $180°$, $270°$ rotations, and horizontal and vertical flips. Then we calculated the perceptual hashes of all images in the augmented train split, identified exact and augmented duplicates by detecting hash collisions, and retained only truly unique train images. The retained image from each set of duplicates was determined arbitrarily. We

followed the same procedure for the official validation split to obtain a subset of unique images for final validation.

**Removal of Data Leakage.** Finally, for these remaining images, we examined hash collisions between the train and validation splits and eliminated all instances of leaked validation images in the train split.

# 3 TECHNICAL VALIDATION & REPRODUCIBILITY

In our experiments, we used a perceptual hashing algorithm with a bit depth of 64 to compute and store the hashes of each image in a dataset. For hash collision detection, we treated exact hash matches as a collision, i.e., adopted a Hamming distance threshold of 0 between the computed hashes. On average, hash computation for a single image takes around 4ms. Hash comparisons were made by a simple equality check, resulting in a highly efficient fast-to-compute de-duplication/leakage detection workflow for large-scale image datasets. All experiments were conducted on a machine with an AMD Epyc 7313 processor and 32GB of memory.

| Search Set | Targets Set | Total number of images in Search set | Number of targets in Search set | Overlap % |
|---|---|---|---|---|
| Official Train | Official Train | 72,000 | 2 | $2.78 \times 10^{-3}\%$ |
| | Official Test | 72,000 | 6 | $8.33 \times 10^{-3}\%$ |
| Official Test | Official Train | 72,000 | 7 | $9.72 \times 10^{-3}\%$ |
| | Official Test | 72,000 | 4 | $5.56 \times 10^{-3}\%$ |
| Augmented Train | Augmented Train | 72,000 | 12 | $16.67 \times 10^{-3}\%$ |
| | Augmented Test | 432,000 | 58 | $13.42 \times 10^{-3}\%$ |
| Augmented Test | Augmented Train | 432,000 | 86 | $19.91 \times 10^{-3}\%$ |
| | Augmented Test | 72,000 | 18 | $25 \times 10^{-3}\%$ |

Table 2: **Data Leakage and Duplication.** Summary of the extent of data leakage/duplication in the train and test splits of the INRIA dataset. The 'Official' train/test sets are those provided by the INRIA Aerial Image Labelling dataset (Maggiori et al., 2017). The 'Augmented' train/test sets refer to those obtained after augmenting the official sets with 90°, 180°, 270° rotations, and horizontal and vertical flips.

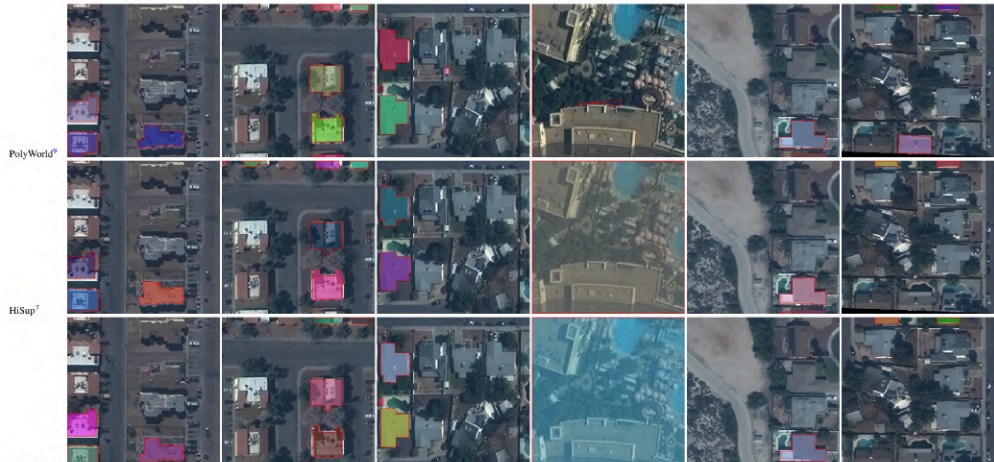

Figure 3: **Qualitative comparisons.** Examples from the original AICrowd (Mohanty et al., 2020) (CC BY-NC-SA 4.0) validation set where images are annotated incorrectly. We show example predictions from PolyWorld (Zorzi et al., 2022) (first row) and HiSup (Xu et al., 2023) (second row). The ground truth is shown in the third row. In these examples, it can be seen that these methods replicate the incorrect/incomplete ground truth annotations, indicating overfitting due to data leakage between the train and validation splits.

## 3.1 Evaluation of INRIA Aerial Image Labelling dataset

We used our de-duplication pipeline to evaluate the quality of the INRIA Aerial Image Labelling dataset (Maggiori et al., 2017). The results of these evaluations are presented in Table 2.

From Table 2, it can be seen that there is negligible data leakage or duplication in the official splits of the INRIA Aerial Image Labelling dataset (Maggiori et al., 2017). Furthermore, the detected leaked samples were simply low-contrast images containing only water bodies or grasslands with little to no buildings. Therefore, these detected leaked samples could be treated as false positives, indicating that there is no real data leakage or duplication in the officially provided training and test splits.

In Figure 7, we depict some qualitative examples of detected data leakage instances across the official train and test splits of the INRIA Aerial Image Labelling dataset (Maggiori et al., 2017). From Figure 7, it can be seen that the hashing technique detects leakage instances despite these patches being from different geographical locations. This is because the hashing technique is invariant to color and small structural changes. The technique can be made more sensitive to smaller structural changes by increasing the bit depth of the hashing algorithm, however, we found that a bit depth of 64 was sufficient for the scope of this study.

| Search Set | Targets Set | Total number of images in Search set | Number of targets in Search set | Overlap % |
|---|---|---|---|---|
| Official Train | Official Train | 10,593 | 105 | 0.991% |
| | Official Test | 10,593 | 96 | 0.906% |
| Official Test | Official Train | 3,526 | 43 | 1.219% |
| | Official Test | 3,526 | 30 | 0.851% |
| Augmented Train | Augmented Train | 10,593 | 163 | 1.539% |
| | Augmented Test | 63,558 | 708 | 1.114% |
| Augmented Test | Augmented Train | 21,156 | 313 | 1.479% |
| | Augmented Test | 3,526 | 43 | 1.219% |

Table 3: **Data Leakage and Duplication.** Summary of the extent of data leakage/duplication in the train and test splits of the SpaceNet v2 dataset. The 'Official' train/test sets are those provided by the SpaceNet v2 dataset (Etten et al., 2018). The 'Augmented' train/test sets refer to those obtained after augmenting the official sets with 90°, 180°, 270° rotations, and horizontal and vertical flips.

| Search Set | Targets Set | Total # of images in Search set | Duplicates detected using PHash | Duplicates detected using AHash |
|---|---|---|---|---|
| Official Train | Official Train | | 166,193 | 167,829 |
| | Official Val | 280,741 | 95,241 | 97,950 |
| | Official Test | | 95,884 | 98,519 |
| Official Val | Official Train | | 56,368 | 56,431 |
| | Official Val | 60,317 | 10,658 | 11,225 |
| | Official Test | | 20,642 | 21,204 |
| Official Test | Official Train | | 56,608 | 56,740 |
| | Official Val | 60,697 | 20,639 | 21,293 |
| | Official Test | | 10,701 | 11,338 |

Table 4: **Comparison of Duplicates and Leakage Detection Using Perceptual and Average Hashing Techniques.** Summary of the extent of data leakage/duplication in the official train, validation, and test splits of the AICrowd dataset (Mohanty et al., 2020). It can be seen that the presence of data leakage and duplication in the AICrowd dataset is confirmed by both perceptual hashing (PHash) and average hashing (AHash) approaches.

## 3.2 Evaluation of SpaceNet 2: Building Detection v2 dataset

The results of the duplication and data leakage evaluations conducted on the SpaceNet 2: Building Detection v2 dataset (Etten et al., 2018) are presented in Table 3. From Table 3, it can be seen that the SpaceNet 2 dataset also exhibits negligible data leakage and duplication. In this case, as well, the detected leaked/duplicate samples were simply no data rasters, which are a common artifact of georeferenced satellite imagery. Therefore, these could also be considered false positives, indicating there is no real data duplication/leakage in the SpaceNet 2 dataset.

In Figure 8, we depict some qualitative examples of detected instances of leakage between the official train and test splits of the SpaceNet 2: Building Detection v2 dataset (Etten et al., 2018). It can be

seen that all of the detected instances of leakage are due to the 'no data' regions. Although there are very minor differences between the detected instances of leakage, the hashing algorithm is invariant to such minor structural differences.

### 3.3 Evaluation of AICrowd Mapping Challenge dataset

**Data Leakage between train, validation, and test splits of the AICrowd dataset:** Initial comparisons indicated that 93.45% (56,368) of the 60,317 official validation images were also present in the training split. In contrast, 33.92% (95,241) of the 280,741 official training images were exact duplicates of images in the validation split. Furthermore, 93.26% (56,608) of the 60,697 official test split images were also present in the training split. The results of these experiments are presented in Table 1 and some examples are illustrated in Figures 2, 5, and 6.

**De-duplication of the AICrowd dataset:** After removing duplicates and augmented duplicates from the official train and validation splits of the AICrowd dataset, the train split contained 29,338 unique images (out of the original 280,741) and the validation split contained 14,166 unique images (out of the original 60,317). From this subset, instances of leakage of validation images in the training split were identified and removed to further prune the training split to 15,392 images. This demonstrates that the AICrowd dataset exhibits severe redundancy and duplication of images.

**Overfitting exhibited by methods reporting on the AICrowd dataset:** Due to the presence of substantial duplication and data leakage in the official splits of the AICrowd dataset, it was discovered that several recently reported methods exhibit severe overfitting. This is particularly evident when these methods even replicate incorrect ground truth annotations from the training dataset. Qualitative examples of this behavior are shown in Figure 3. This explains why these methods achieve such high evaluation scores on the dataset.

### 3.4 Comparison of the Perceptual Hashing Pipeline with Average Hashing

To reasonably verify the results of the analyses conducted using the perceptual hashing pipeline, we also ran checks using a standard average hashing algorithm for detecting data leakage and duplicates across the official train, validation, and test splits of the AICrowd dataset (Mohanty et al., 2020). The results of these comparisons are presented in Table 4. Therefore, it can be seen that analyses using both perceptual hashing and average hashing detect a similar extent of data leakage and duplication in the official train, validation, & test splits of the AICrowd dataset. This confirms that the AICrowd dataset suffers from considerable data leakage and duplication issues. The minor difference in the detected number of duplicates is because average hashing is prone to false positives, where similar images are sometimes incorrectly flagged as duplicates, as shown in Figure 4. From these results, it is clear that perceptual hashing is less prone to false positive errors and hence is a more suitable choice for evaluating large-scale image datasets.

### 3.5 Discussion

From the results presented above, it can be observed that the deduplication pipeline is effective at detecting instances of duplication and leakage in large image datasets. The issues of leakage and duplication discovered in the AICrowd Mapping Challenge dataset render it unsuitable for benchmarking building footprint extraction methods without removing the leakage and duplication instances. This also potentially invalidates the quantitative metrics reported on this dataset by several preceding works. We also observe that the INRIA Aerial Image Labelling and SpaceNet 2: Building Detection v2 datasets are generally devoid of such major issues and could serve as more suitable datasets for benchmarking future research focusing on the task of building footprint detection.

**Experimental details:** The hash computations and comparisons for all experiments were conducted on a machine with an AMD EPYC 7313 server-grade CPU after allocating 8 cores for the jobs. The experiments revealed average runtimes of 4ms per image for hash computation and 4ms per comparison for hash comparisons.

**Choice of Datasets:** The objective of this study was to evaluate the quality of the most common and popular geospatial datasets in the building footprint extraction literature. However, the proposed

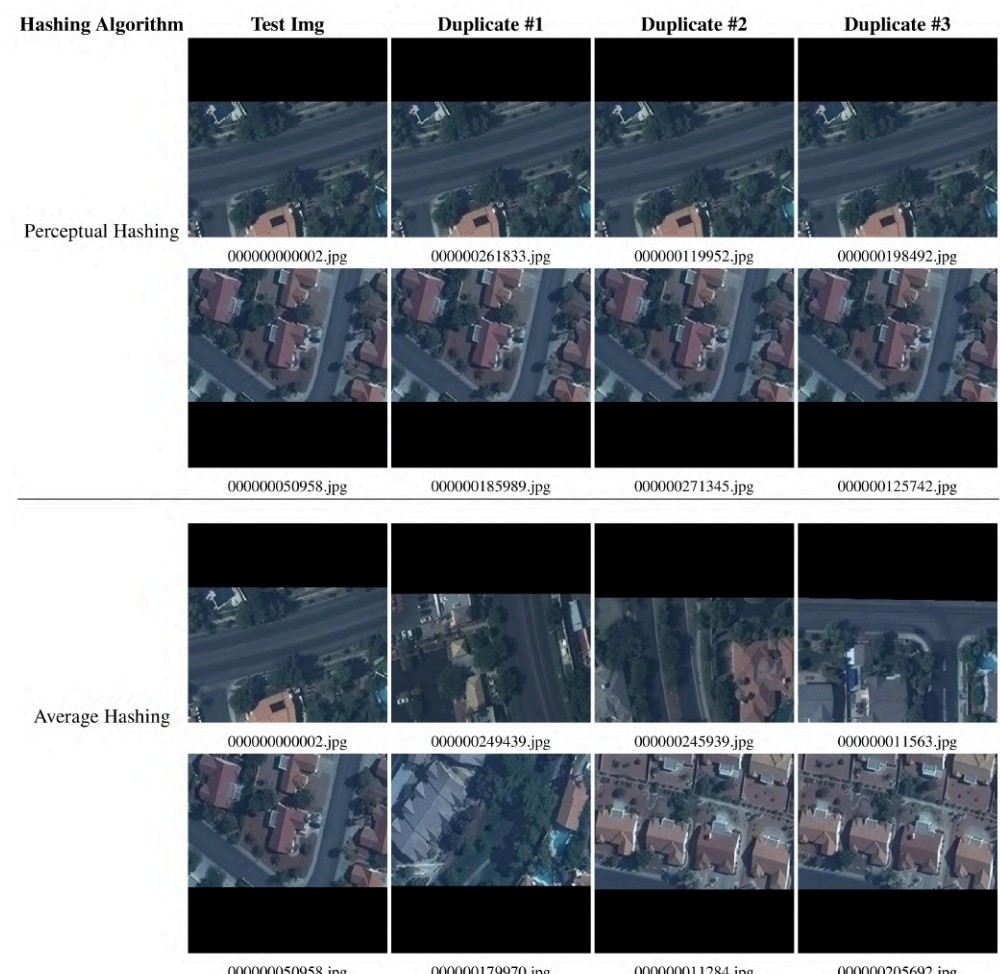

Figure 4: **Qualitative Comparisons of Duplicates detected using Perceptual Hashing vs. Average Hashing.** Here we show examples of data leakage in the AICrowd dataset (Mohanty et al., 2020) (CC BY-NC-SA 4.0). We sample two images from the **test split** in the $1^{st}$ column and show duplicates occurring in the **training split** in the $2^{nd}$, $3^{rd}$, and $4^{th}$ columns. It can be seen that the Perceptual Hashing approach is less prone to false positives when compared to the Average Hashing approach.

deduplication pipeline can be easily used for the assessment of any large-scale image dataset such as ImageNet (Deng et al., 2009), VOC (Everingham et al., 2010), MS-COCO (Lin et al., 2014), Cityscapes (Cordts et al., 2016), etc.

**Limitations:** Despite the effectiveness of the proposed pipeline, there are also some limitations worth noting. In the present pipeline, although the perceptual hashing algorithm is invariant to radiometric augmentations (such as brightness & contrast changes), it is not inherently invariant to geometric augmentations such as rotation or flips. We overcome this limitation by augmenting the input images before the hash computation as part of the pipeline. The augmentations were chosen based on an initial visual inspection of the nature of duplications occurring in the datasets. Therefore, the choice of augmentations depends on the statistics of the dataset, i.e., some a priori information about the dataset is required before choosing appropriate augmentations. These limitations could be addressed in future research by developing more robust hashing algorithms that are inherently invariant to strong geometric and radiometric transformations.

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

## A  APPENDIX

In this appendix, we show qualitative examples of data leakage and duplication discovered in the AICrowd Mapping Challenge dataset (Mohanty et al., 2020) in Figures 5 and 6. Additionally, in the case of the INRIA Aerial Image Labelling Dataset (Maggiori et al., 2017) and the SpaceNet 2 Building Detection v2 dataset (Etten et al., 2018), we also depict some examples of the false positive duplicates identified by the deduplication pipeline in Figures 7 and 8 respectively.

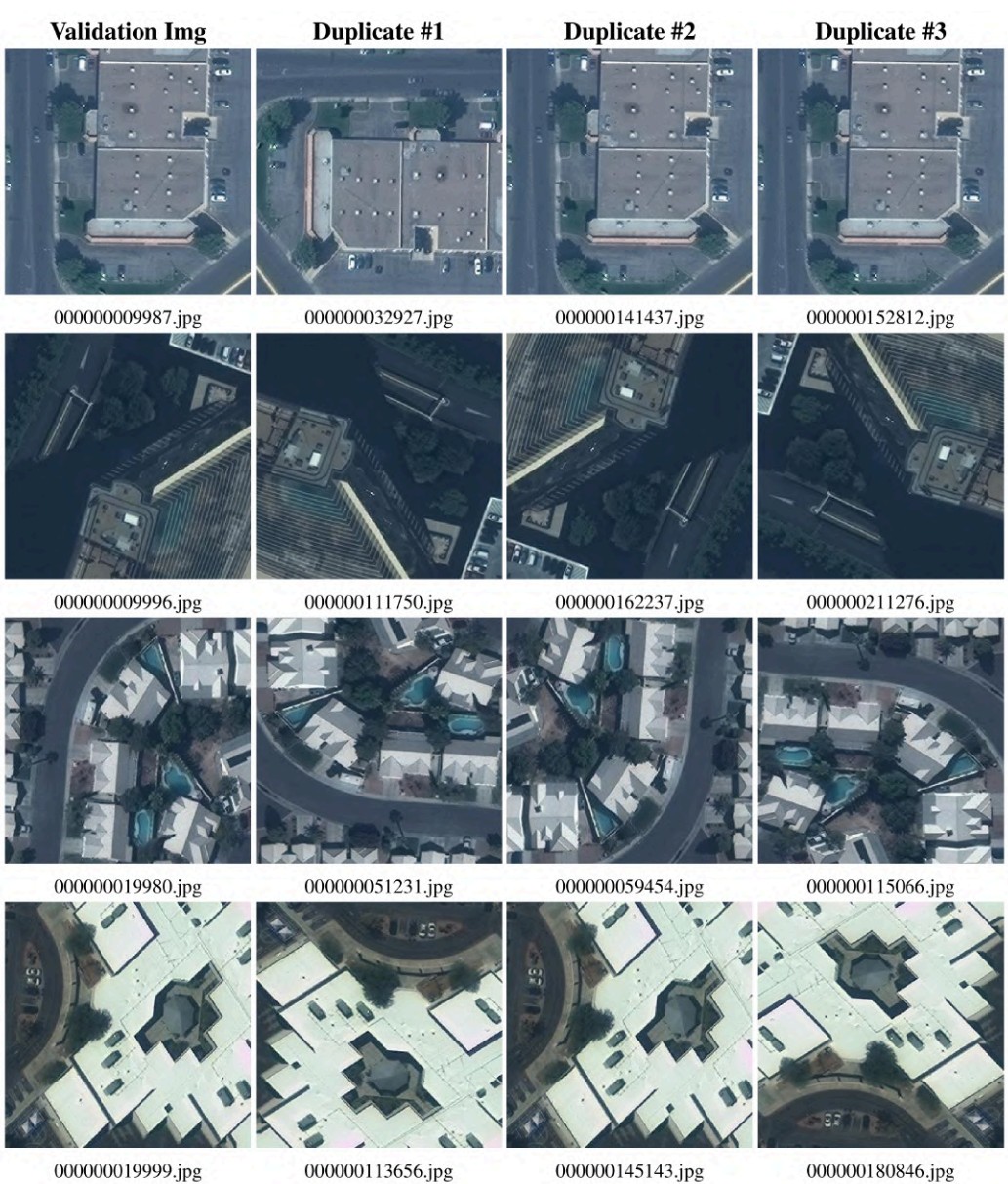

Figure 5: **Additional examples of data leakage.** Here we show additional examples of data leakage in the AICrowd Mapping Challenge dataset (Mohanty et al., 2020) (CC BY-NC-SA 4.0). We sample four images from the **validation split** in column 1 and show duplicates occurring in the **training split** in columns 2, 3, and 4.

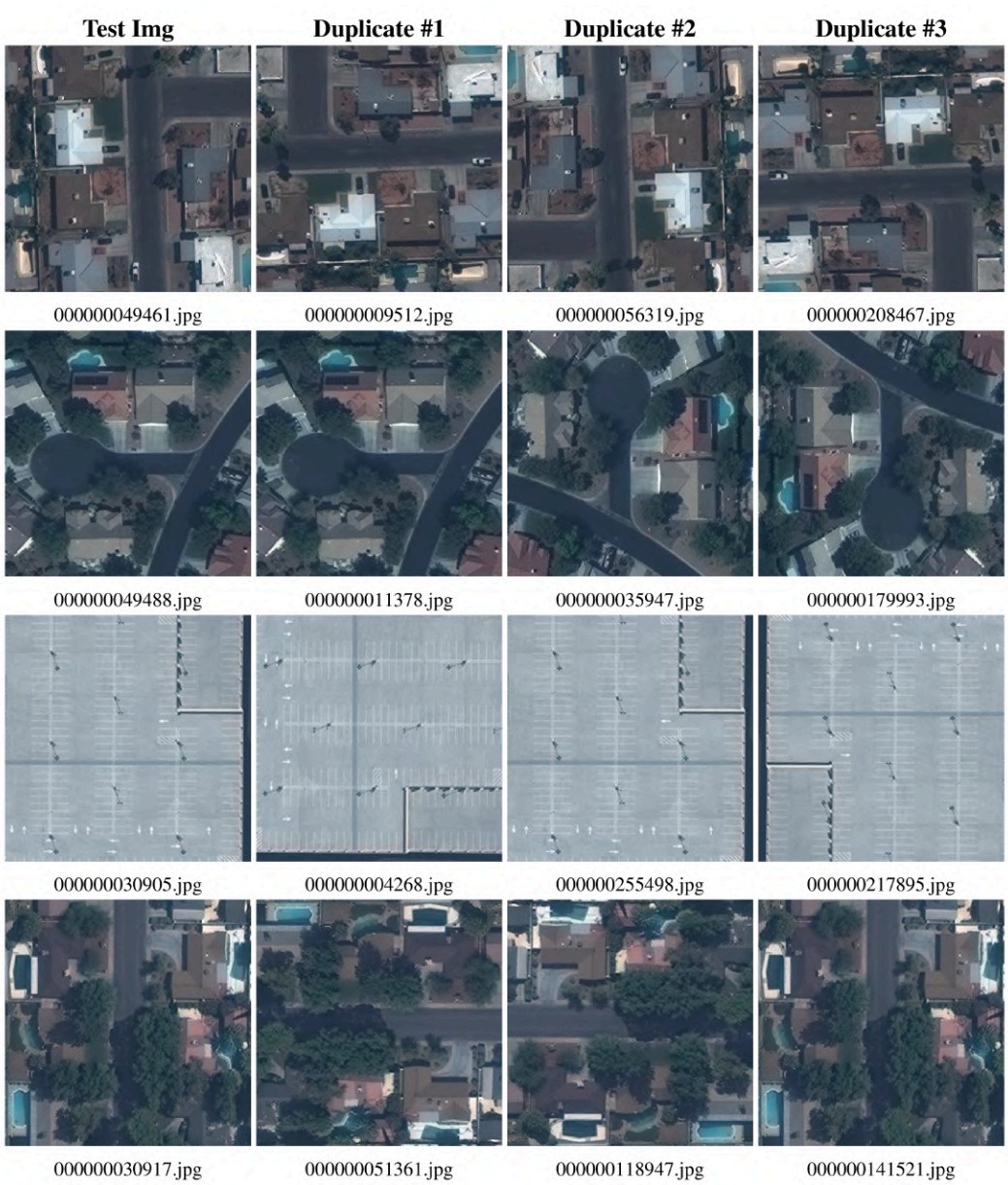

Figure 6: **Additional examples of data leakage.** Here we show additional examples of data leakage in the AICrowd Mapping Challenge dataset (Mohanty et al., 2020) (CC BY-NC-SA 4.0). We sample four images from the **test split** in column 1 and show duplicates occurring in the **training split** in columns 2, 3, and 4.

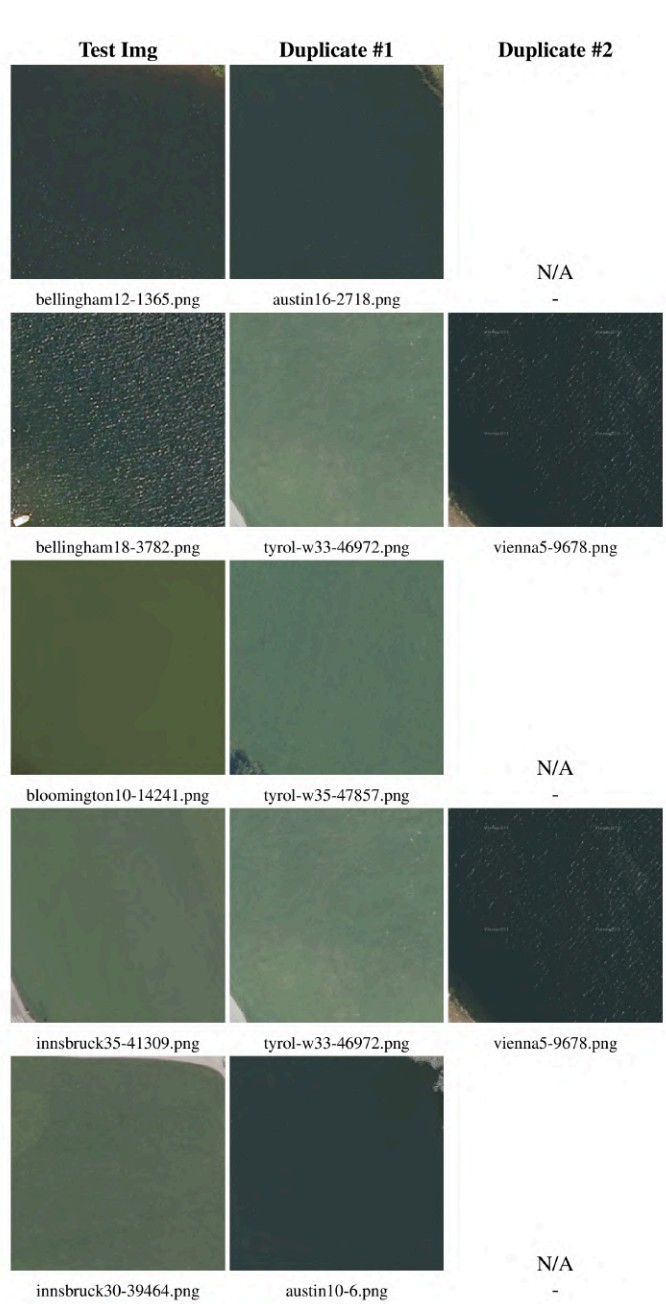

Figure 7: **False positive examples of data leakage.** Here we show falsely detected examples of data leakage in the INRIA Aerial Image Labelling dataset (Maggiori et al., 2017). We sample images from the **test split** in column 1 and show duplicates occurring in the **training split** in columns 2 and 3.

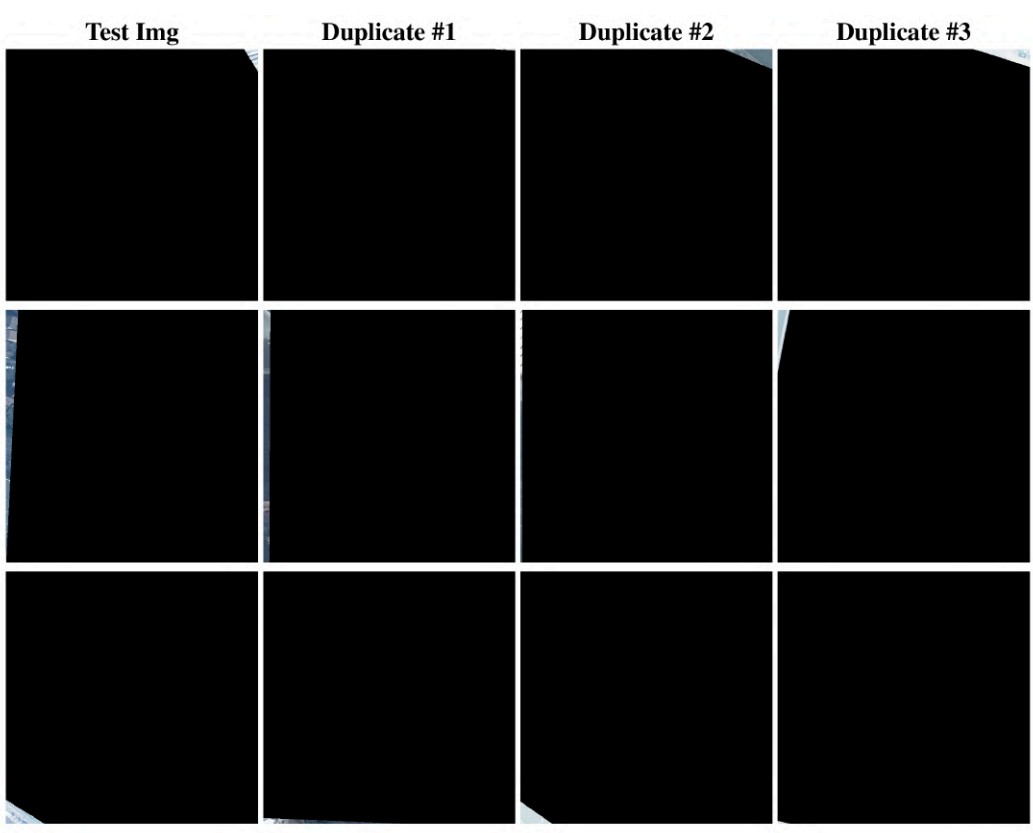

Figure 8: **False positive examples of data leakage.** Here we show examples of falsely detected examples of data leakage in the SpaceNet 2: Building Detection v2 dataset (Etten et al., 2018) (CC BY-SA 4.0). We sample four images from the **test split** in column 1 and show duplicates occurring in the **training split** in columns 2, 3, and 4.

