# OpenReview forum: "Data Leakage Detection and De-duplication in Large Scale Geospatial Image Datasets"
_ICLR.cc/2026/Conference — ICLR 2026 Conference Withdrawn Submission_

### Official Review · Reviewer_nGTg · 2025-10-26

**Soundness:** 2
**Presentation:** 1
**Contribution:** 2
**Rating:** 2
**Confidence:** 4

**Summary:**

This paper conducts a data quality analysis of three remote sensing image datasets — INRIA, SpaceNet 2, and AICrowd. Its core contribution lies in revealing the severe data leakage present in the AICrowd Mapping Challenge dataset. The authors propose a data validation pipeline based on perceptual hashing techniques, which enables the efficient identification of data duplication and leakage.

**Strengths:**

（1）This study exposes catastrophic data contamination in the widely used benchmark dataset AICrowd Mapping Challenge, which holds significant corrective value for the entire subfield of building footprint extraction.

（2）The proposed pipeline demonstrates high generalizability and can serve as a lightweight tool for pre-screening large-scale image datasets, providing a useful engineering baseline for subsequent research.

**Weaknesses:**

(1) Lack of methodological novelty.

The proposed pipeline is essentially a reimplementation of the standard perceptual hashing (pHash) algorithm without introducing new algorithmic contributions or theoretical insights.

(2) No validation after dataset cleaning.

The paper only exposes the data-leakage issue but does not retrain models or provide quantitative evidence showing that de-duplication improves generalization or benchmark reliability.

(3) Limited scope and poor writing quality.

Experiments are restricted to three building-footprint datasets with no verification on larger benchmarks. The overall writing and figure presentation quality are low, with rough illustrations and limited clarity, which undermines the readability and professionalism of the paper.

**Questions:**

1. How sensitive are the results to the choice of the Hamming distance threshold in hash matching? Even a small tolerance (e.g., ≤3 bits) could alter the duplication count—was this sensitivity analyzed?

2. Can the authors provide quantitative results comparing model performance before and after dataset cleaning to confirm that removing duplicates truly mitigates overfitting and improves robustness?

---

### Official Review · Reviewer_HpfA · 2025-10-31

**Soundness:** 2
**Presentation:** 2
**Contribution:** 1
**Rating:** 2
**Confidence:** 4

**Summary:**

The paper analyzes three popular remote-sensing building-footprint datasets—INRIA Aerial, SpaceNet 2, and especially the AICrowd Mapping Challenge—and shows that AICrowd suffers from severe data leakage, with about 90% of its training images and about 93% of its validation images duplicated or nearly duplicated across splits, and even test images appearing in training, which inflates benchmark results and can make models simply memorize flawed labels rather than learn building extraction. To address this, the authors propose a fast, scalable perceptual-hashing pipeline based on low-frequency DCT features to detect exact and lightly augmented duplicates both within and across splits, and after cleaning they find the ostensibly large AICrowd training set collapses to only about 15k unique images, revealing how misleading the original dataset size was. They conclude that geospatial vision datasets, which often reuse similar tiles and augmentations, should always be de-duplicated before training or leaderboard evaluation, and that prior results on contaminated splits should be treated with caution.

**Strengths:**

- The findings in this work are important for the quality of benchmarking in the remote sensing community

**Weaknesses:**

- There is very limited novelty to the proposed perceptual hashing algorithm, in fact based on the description they seem nearly identical to the ones used in public libraries (https://github.com/JohannesBuchner/imagehash)

- Though the authors identify an issue with the AICrowd benchmark, they do not clean it up, nor do they re-evaluate existing methods on a new cleaned up version of the dataset

- There are a variety of other remote sensing datasets which could potentially be analyzed in this manner, e.g. DOTA, XVIEW etc.

- Overall the value is limited to making the community aware of the issue of data leakage in this specific dataset

**Questions:**

- Aerial data often has near-duplicates from the same location but different time (seasonal change, lighting, construction) that you might actually want to keep, how do you account for this ?

---

### Official Review · Reviewer_8XQY · 2025-11-01

**Soundness:** 2
**Presentation:** 2
**Contribution:** 2
**Rating:** 4
**Confidence:** 2

**Summary:**

This paper audits duplication and cross-split leakage in large geospatial image datasets and proposes a lightweight, scalable de-duplication pipeline based on 64-bit perceptual hashing with simple geometric augmentations to catch rotated/flipped duplicates efficiently. The authors evaluate three public benchmarks, finding negligible true leakage in INRIA and SpaceNet v2, but severe contamination in AICrowd, where large fractions of validation/test images reappear in training, including augmented overlaps. They also compare perceptual hashing against average hashing, showing fewer false positives with the former while both corroborate the extent of AICrowd contamination.

**Strengths:**

- The paper uncovers severe, decision-relevant data defects.

- It delivers a practical, scalable pipeline that processes large datasets efficiently.

- Qualitative comparisons show perceptual hashing is less prone to false positives than average hashing while still corroborating the extent of contamination, supporting the chosen technique.

**Weaknesses:**

- Baseline coverage is minimal. The study does not establish competitiveness against a broader set of leakage detectors.

- There is no ablation or systematic sensitivity analysis of key design choices.

- The evaluation does not quantify false positive or false negative rates with statistical tests.

- Treating hash matches with a strict Hamming distance of 0 risks false negatives for near-duplicates that differ by small artifacts, and the paper does not explore tolerance thresholds.

- When duplicates are found, the choice of which image to keep is made arbitrarily, which could bias any downstream analyses that depend on image selection.

- The paper lacks formal theoretical analysis and instead relies mainly on descriptive algorithms and empirical runtimes.

**Questions:**

- How sensitive are results to hash bit-depth, augmentation set, and candidate selection rules?

- What are the estimated false-positive and false-negative rates, and are they statistically distinguishable across methods?

- Did you explore small Hamming radii or bit-error tolerance to recover near-duplicates with minor artifacts? What FP increase did you observe?

- How do you decide which image to keep when duplicates are found, and could that choice bias downstream tasks?

- Can you formalize the time and space complexity of each stage?

- How did you adjudicate ambiguous matches (e.g., very similar but not identical tiles)?

---

### Official Review · Reviewer_X3fA · 2025-11-05

**Soundness:** 2
**Presentation:** 3
**Contribution:** 2
**Rating:** 2
**Confidence:** 5

**Summary:**

The paper introduces an image hashing procedure for the application of duplicate identification in collections of overhead imagery. The algorithm extracts large-scale features by image coarse-graining and low-pass filtering on the spatial modes. The approach is applied to three popular building foodprint extraction Earth observation datasets, namely: INRIA Aerial Image, SpaceNet 2, and the AICrowd Mapping. According to the analysis, the latter ships with major flaws in data duplication for the training data and significant data overlap with the test data.

**Strengths:**

The paper is clearly written in proper English alongside illustrations by figures and tables. The approach is to the point and valuable in regards of quality checks of benchmark datasets employed by the remote sensing community. The paper does not leave the reader with major open questions.

**Weaknesses:**

Despite its clarity and value in general, I rate the scientific contribution to ICLR limited. No extensive experiments with models have been presented, nor does the hasing procedure (Fig. 1) provide relevant novel aspects in regard to representation learning. In fact, the paper would benefit from an ablation and sensitivity analyis of the hash-matching when corresponding hyper-parameter such as downsampling size and low-pass filtering vary. The key message of the work presented is publicly available since early 2023, https://arxiv.org/abs/2304.02296.

**Questions:**

none

---

### Note · Authors · 2025-11-14

**Comment:**

We thank the reviewers for their valuable feedback. Considering the comments received, we would like to withdraw our submission and work on the suggested improvements.

**Withdrawal Confirmation:**

I have read and agree with the venue's withdrawal policy on behalf of myself and my co-authors.